# Disability and educational outcomes in Nepal: Evidence from the 2019 Multiple Indicator Cluster Survey

**Prashant Acharya** [ID] *

Bodh Insight, Kathmandu, Nepal

* 707prashantacharya@gmail.com

## Abstract

There is limited evidence on how disability affects children's educational outcomes in low- and middle-income countries, particularly during adolescence. This study aims to examine the association between disability and various educational outcomes for children ages 5–17 in Nepal. This study uses a nationally representative sample of 7,824 children ages 5–17 surveyed in the 2019 Nepal Multiple Indicator Cluster Survey. In the survey, functional disability was measured as a binary variable (=1 if the respondent reported having a functional disability, 0 otherwise). The key educational outcomes examined are the following: (1) whether the child ever attended school (binary), (2) current grade level (continuous), and (3) whether the child is in the appropriate grade for their age (binary). The association between disability status and the outcomes was examined in a regression framework controlling for age, gender, geographic location, mother's education, and household wealth index. Among 7,790 children with complete data, approximately 13.7% children reported having a functional disability. After adjusting for potential confounders, having a disability was associated with a 2.7 percentage point reduction in the likelihood of ever attending school (p < 0.001). Likewise, having a disability was associated with the completion of 0.21 fewer grades on average (p < 0.001). Having a disability was not associated with reaching the appropriate grade level for age (p = 0.64). Children with disabilities were 13.5 percentage points more likely to report facing difficulties adjusting to routine changes in their daily activities (p < 0.001). These findings highlight persistent educational disadvantages among children with disabilities in Nepal and underscore the need to create inclusive learning environments and support systems to promote equitable access to education and reduce educational inequalities.

## Introduction

Disability affects over a billion individuals globally—around 16% of the world's population—and the burden of disability is higher in low- and middle-income countries

**Data availability statement:** All data are fully available without restriction. The data underlying the results presented in the study are available from the Multiple Indicator Cluster Surveys (MICS) which can be downloaded from https://mics.unicef.org free of charge (requires registration).

**Funding:** The authors received no specific funding for this work.

**Competing interests:** The authors have declared that no competing interests exist.

(LMICs). [1] An estimated 80% of individuals with disabilities live in LMICs, and one in five of the world's poorest people has some form of disability. [2,3] These numbers are expected to rise further as populations age and chronic health conditions become more common. [4] People with disabilities, especially children and adolescents, face greater challenges than their peers without disability in accessing education, healthcare, and employment. [5–8] In many LMICs, policy-level actions aimed at supporting this group remain limited. When inclusive policies exist, they are mostly on paper and their implementation remains poor. [9–11]

Different studies have shown that disability can severely affect children's various educational outcomes. [12–15] Children with disabilities are far less likely to start school or stay enrolled—gaps in attendance between children with and without disabilities are as high as 30 percentage points on average. [14] In some countries, this gap can go up to 50 percentage points. [13] Globally, children with disabilities are less likely to attend school, complete primary or secondary education, and tend to have fewer years of schooling than children without disabilities. [16] Along with that, chronic disability conditions are also significantly associated with increased absenteeism, grade repetition and not completing high school within four years. [15]

There are several barriers to education for children with disabilities in LMICs which includes health issues, lack of family resources, reduced cognitive functioning, stigma, lack of trained teachers, inaccessible transportation, and poor school infrastructure. [16,17] Even though disability may not always be strongly tied to poverty, its long-term impact on educational outcomes can reinforce a cycle of disadvantage. [13]

Mirroring the global scenario, Nepal, the site of the current study, has also experienced a rise in disability prevalence in recent years, although the availability and accuracy of the data remains a concern. The 2021 National Population and Housing Census reports that 2.2% of Nepal's population—roughly 647,744 individuals—live with some form of disability [18], up from 1.9% in the 2011 census. Among them, 54.2% are male and 45.8% are female. Physical disabilities account for the largest portion (37.1%), followed by low vision (17.1%), hearing impairments (15.9%), blindness (5.4%), and other types of disabilities which includes intellectual and psychosocial factors. [18]

Several studies have highlighted the barriers faced by children with disabilities in accessing education within the context of Nepal. [8,19] Children with visual and hearing impairments often encounter inadequate support systems in schools, while those with physical impairments face challenges related to inaccessible infrastructure. [19] Common barriers such as financial constraints, parental attitudes, and social stigma further hinder educational access for these children. [19] Another study explored the situation of educating students with intellectual disabilities in Nepal, revealing that despite efforts, such as special education services and community-based instructions, challenges like inadequate teacher training and insufficient resources persist. [20] Similarly, issues such as the parental lack of awareness and negative attitudes, and policy implementation gaps deter school transition for children with disabilities. [21]

On the policy front, over the years, Nepal has introduced new laws to improve inclusion of individuals with disabilities. The Disability Rights Act 2017 replaced the earlier Disabled and Protection Welfare Act 1982 and includes better provisions around education, employment, healthcare, and mobility. [22] Despite that, many children with disabilities still do not have access to quality education.

In Nepal, children typically begin formal schooling at age five with entry into Grade 1. Basic education is compulsory and covers Grades 1–8, followed by secondary education from Grades 9–12 [23]. Over the past two decades, Nepal has made substantial progress in expanding school access, with net enrollment rates for grades 1–5 increasing from around 80 percent in the early 2000s to over 95 percent [23]. However, challenges related to grade progression, repetition (reaching up to 12% in early grades), and dropout (around 4–7% annually) persist, particularly among children from disadvantaged cast/ethnic groups [24]. Disability intersects with these structural features of the education system, contributing to disparities in school participation and educational attainment.

The objective of the current study is to examine how functional disability relates to educational outcomes among children ages 5–17 in Nepal using nationally representative data from the 2019 Multiple Indicator Cluster Survey (MICS). The broader objective is to contribute to policy discussions that aim to promote inclusive education and support equal opportunities for all children—regardless of their abilities.

## Methodology

We use data from the 2019 Nepal MICS. The MICS dataset, developed through collaboration between the Central Bureau of Statistics (CBS) and UNICEF, is nationally representative and includes comprehensive information on 7,790 children ages 5–17 years. It provides detailed demographic, economic, and educational information, making it suitable for evaluating how socioeconomic conditions influence educational access and achievement.

The dataset was accessed on 12 April 2024 through the UNICEF MICS data portal. The MICS data are fully anonymized, and no personally identifiable information was available to the researchers.

The educational outcomes in this study are assessed using four dependent variables: (1) whether the child has ever attended school or an early childhood education program (binary), (2) the highest grade level currently attended (continuous), (3) whether the child is currently enrolled in an age-appropriate grade (binary), and (4) difficulty adjusting to routine changes (binary) based on the mother's or primary caregiver's report of whether the child has trouble adapting to changes in general daily activities. The fourth outcome captures psychological and developmental adaptability.

The key independent variable in this study is whether an individual has functional disability. Each respondent (parent of the children) was asked different questions about some functional disabilities the child may have. The questions covered various aspects of functional disabilities such as difficulties in hearing, seeing, walking, learning and behavioral problems that the child could be facing. If the child was facing any of these difficulties, s/he was categorized as functionally disabled.

Several control variables are used in the regression analysis to account for potential confounding. These include gender, age, province, geographic location (rural vs urban), wealth index, and mother's education level. These covariates were selected based on prior research indicating their potential influence on educational access and performance. Age was grouped into 5–9, 10–14, and 15–17 years to reflect key schooling transitions and dropout risks in Nepal. Children 5–9 years are in the early primary grades, while ages 10–14 years span later primary and entry into lower secondary level, when progression starts to diverge. [25,26] The 15–17 age group corresponds to upper secondary schooling, during which risks of delayed progression and school dropout increase, particularly due to factors such as early marriage. [26–28] Data cleaning, preprocessing and all other statistical analysis were performed using Stata version 16.0.

The study uses multivariate regression model to assess the relationship between disability and educational outcome. Robust standard errors are reported for all models, and statistical significance is noted where appropriate. Following standard practice in social science research, we report significance levels up to 10 percent.

The general regression specification is as follows:

$$Y_i = \beta_0 + \beta_1 \text{Disability}_i + \beta_2 \text{Age}_i + \beta_3 \text{Female}_i + \beta_4 \text{Province}_i + \beta_5 \text{Rural}_i + \beta_6 \text{Mother's Education}_i + \varepsilon_i$$

where $Y_i$ is the educational outcome for child *i*, and $\varepsilon_i$ is the error term. This model allows us to estimate the association between disability and educational outcomes while adjusting for other important factors. While some outcomes are binary (e.g., school attendance and grade-for-age alignment), we use linear probability models for all outcomes for the ease of interpreting the coefficients and comparing the associations across studies. This choice also avoids common confusion around odds ratios, which can differ substantially from relative risks when the outcome is common. Using OLS makes the results easier to interpret and compare across models. [29]

In addition to the main analysis, we also examine if the association between disability and the educational outcomes varies across two key dimensions of potential differences: gender and area of residence. We focus on these two dimensions because disparities in health and educational outcomes by gender and area of residence in Nepal have been widely documented in the existing literature [30,31], and it is possible that those differences persist vis-à-vis disability status. For this examination, we estimate regressions similar to the one in the main analysis by including an interaction term between disability and gender and, separately, between disability and urban versus rural residence. After conducting the regressions, we generate graphs showing the marginal effects.

## Findings

The descriptive statistics in Table 1 provide an overview of how functional disability is distributed across demographic and socioeconomic characteristics. Among the 7,790 children in the sample, approximately 13.66 percent were reported to have some form of functional disability. The prevalence was notably higher among younger children: nearly half of the children with disabilities (49.53 percent) were between ages 5–9, compared to only 16.64 percent in the 15–17 age group. Children residing in urban areas made up a slightly larger share of the population with disabilities (55.73 percent) compared to those in rural areas (44.27 percent).

There were also notable differences across wealth groups. Almost 30 percent of children with disabilities came from the poorest wealth quintile, while just over 11 percent belonged to the richest quintile. A slightly higher proportion of children with disabilities were male (53.95 percent) compared to female (46.05 percent). These differences suggest socioeconomic and geographic patterns in the reporting or experience of disability.

In terms of educational indicators, children with disabilities were less likely to have participated in early childhood education programs. Among children with disabilities, 95.3 percent had ever attended school or an early childhood program, compared to 98.2 percent of children without disabilities. The gaps were even wider in adaptability. While 85.91 percent of children without disabilities had no difficulty adjusting to changes in routine, only 71.52 percent of children with disabilities fell in this category. Over 26 percent of children with disabilities were reported to experience either some difficulty or high difficulty in accepting changes, compared to around 14 percent among their non-disabled peers.

Table 2 presents the results of multivariate regression analyses examining the relationship between functional disability and key educational outcomes while controlling for age, gender, rural residence, province, mother's education, and household wealth quintile. Functional disability is significantly associated with a lower likelihood of ever attending school or an early childhood education program. The coefficient for disability is –0.027 (p < 0.01), indicating that children with functional disabilities are about 2.7 percentage points less likely to attend school compared to those without disabilities. This indicates that for every 100 children, approximately three fewer children with disabilities attend school compared to their peers without disabilities.

Similarly, in terms of academic progression, disability is associated with lower grade attainment. Children with disabilities are enrolled in lower grades compared to their peers, with a significant coefficient of –0.213 (p < 0.01) for highest

**Table 1. Summary Statistics for the Sample.**

| Variables | Sample (n = 7,790) | % | With disability (%) | Without disability (%) |
|---|---|---|---|---|
| **Functional Disability** | | | | |
| Yes | 1,064 | 13.66 | – | – |
| No | 6,726 | 86.34 | – | – |
| **Age** | | | | |
| 5-9 | 3,346 | 42.95 | 49.53 | 41.91 |
| 10-14 | 2,856 | 36.66 | 33.83 | 37.11 |
| 15-17 | 1,588 | 20.39 | 16.64 | 20.98 |
| **Province** | | | | |
| Province 1 | 1,107 | 14.15 | 15.13 | 13.99 |
| Province 2 | 1,220 | 15.59 | 15.6 | 15.58 |
| Province 3 | 1,431 | 18.29 | 17.86 | 18.39 |
| Province 4 | 836 | 10.69 | 11.09 | 10.63 |
| Province 5 | 1,202 | 15.36 | 15.601 | 15.3 |
| Province 6 | 1,001 | 12.79 | 10.62 | 13.1 |
| Province 7 | 1,027 | 13.13 | 14.1 | 13.01 |
| **Area** | | | | |
| Urban | 4,422 | 56.52 | 55.73 | 56.74 |
| Rural | 3,402 | 43.48 | 44.27 | 43.26 |
| **Wealth Index Quintile** | | | | |
| Poorest | 2,259 | 29.00 | 29.98 | 28.84 |
| Second | 1,549 | 19.88 | 21.33 | 19.66 |
| Middle | 1,546 | 19.85 | 22.84 | 19.37 |
| Fourth | 1,384 | 17.77 | 14.76 | 18.24 |
| Richest | 1,052 | 13.50 | 11.09 | 13.89 |
| **Gender** | | | | |
| Male | 3,934 | 50.50 | 53.95 | 49.96 |
| Female | 3,856 | 49.50 | 46.05 | 50.04 |
| **Attended early childhood program** | | | | |
| Yes | 7,618 | 97.79 | 95.3 | 98.19 |
| No | 172 | 2.21 | 4.70 | 1.81 |
| **Education** | | | | |
| None | 1,041 | 13.36 | 18.23 | 12.59 |
| Basic | 5,246 | 67.34 | 68.23 | 67.2 |
| Secondary | 1,492 | 19.15 | 13.35 | 20.07 |
| Higher | 10 | 0.13 | 0.09 | 0.13 |
| **Completed highest grade** | 260 | 3.85 | 3.22 | 3.95 |
| **Difficulty in accepting changes in routine** | | | | |
| No difficulty | 6,539 | 83.94 | 71.52 | 85.91 |
| Some difficulty | 1,180 | 15.15 | 21.9 | 14.08 |
| High difficulty | 51 | 0.65 | 4.79 | 0 |
| Cannot accept at all | 19 | 0.24 | 1.79 | 0 |

Note: Percentages in the 'With disability' and 'Without disability' columns represent the distribution within each subgroup. For instance, 49.53% of children with disability are ages 5–9. These percentages should not be interpreted as the share of children in each demographic group who have a disability.

**Table 2. Results of regression analysis.**

| Variables | Ever Attended School / Program | Highest Grade Attainment | Grade-For-Age Alignment | Difficulties adjusting to Routine |
|---|---|---|---|---|
| Functional Disability | -0.027*** (0.005) | -0.213*** (0.062) | -0.007 (0.016) | 0.135*** (0.012) |
| **Age (ref: 5–9)** | | | | |
| 10-14 | 0.017*** (0.004) | 4.503*** (0.048) | -0.162*** (0.012) | -0.059*** (0.009) |
| 15-17 | 0.006 (0.005) | 7.558*** (0.059) | -0.288*** (0.016) | -0.104*** (0.011) |
| **Province (ref:Province 1)** | | | | |
| Province 2 | -0.058*** (0.006) | -0.999*** (0.081) | -0.230*** (0.021) | -0.041*** (0.016) |
| Province 3 | 0.002 (0.006) | 0.047 (0.077) | 0.024 (0.020) | 0.003 (0.015) |
| Province 4 | 0.007 (0.007) | 0.312*** (0.085) | 0.057*** (0.022) | 0.040** (0.017) |
| Province 5 | -0.007 (0.006) | -0.322*** (0.077) | -0.114*** (0.020) | -0.054*** (0.015) |
| Province 6 | 0.009 (0.007) | 0.075 (0.086) | -0.019 (0.022) | -0.069** (0.017) |
| Province 7 | 0.011* (0.006) | -0.162** (0.081) | -0.127*** (0.021) | -0.094*** (0.016) |
| **Gender (ref:Male)** | | | | |
| Female | -0.003 (0.003) | 0.139*** (0.042) | 0.046*** (0.011) | -0.011 (0.008) |
| **Area (ref:Urban)** | | | | |
| Rural Area | 0.001 (0.004) | 0.080* (0.047) | 0.027** (0.012) | 0.022** (0.009) |
| **Mother's education (ref:no education)** | | | | |
| Has some level of education | 0.018*** (0.004) | 0.275*** (0.048) | 0.108*** (0.012) | 0.001 (0.009) |
| **Wealth Index Quintile (ref:poorest quintile)** | | | | |
| Second | 0.015*** (0.005) | 0.285*** (0.067) | 0.027 (0.018) | -0.023* (0.013) |
| Middle | 0.015*** (0.005) | 0.115 (0.070) | -0.000 (0.018) | -0.027** (0.014) |
| Fourth | 0.028*** (0.006) | 0.252*** (0.074) | 0.050** (0.019) | -0.041*** (0.014) |
| Richest | 0.024*** (0.007) | 0.476*** (0.088) | 0.138*** (0.023) | -0.048*** (0.017) |
| **Constant** | 0.961 | -4.134 | 0.924 | 0.309 |
| **$R^2$** | 0.040 | 0.728 | 0.137 | 0.043 |
| **F-statistic** | 20.272*** | 1273.525*** | 71.018*** | 21.847*** |
| **Sample Size** | 7,789 | 7,617 | 7,185 | 7,789 |

**Note:** This table shows the coefficients from the regression shown in equation (1). The standard errors are shown in parentheses. *$p < 0.10$, **$p < 0.05$, ***$p < 0.01$.

grade attainment. This implies that a child with disability is expected to be about one-fifth of a grade behind a child without a disability. While this gap may seem small, it is important to recognize that even early differences in grade progression can widen over time and contribute to lower overall educational achievement.

We do not find a statistically significance association between disability and being at the appropriate grade level (coefficient = −0.007, p = 0.640), suggesting that, although children with disabilities attend fewer years of school on average, they are not necessarily more likely to be in the "wrong" grade for their age once enrolled. This points to the possibility that, conditional on attending school, many children with disabilities are progressing through grades at a pace similar to their peers.

The analysis also shows that disability is strongly associated with greater difficulty in adapting to the changes in routine. The coefficient is 0.135 (p < 0.01), indicating that children with disabilities are more likely to face behavioral and developmental challenges that could affect their adjustment to structured school environments. Specifically, a child with any form of disability is about 13.5 percentage points more likely to experience some difficulty in adapting to daily routine changes compared to a non-disabled child. These challenges may not directly impact enrollment or grade progression but could significantly affect children's learning experiences and social integration within classrooms.

To assess potential heterogeneity in the association between disability status and educational outcomes, we examined differential associations by gender and area of residence. S1 Table and S1 Fig in Supporting Information section shows that, across all four outcomes—ever attended school, highest grade attained, age-appropriate grade attainment, and difficulty adjusting to routine changes—the association between disability and gender and between disability and area of residence did not differ.

## Discussion

In this study, we examined the association between disability and various measures of education in Nepal. We find that children with disabilities are significantly less likely to attend school and, when enrolled, attain lower grades on average than their non-disabled peers. The data also reveals that children with disabilities are more likely to experience difficulty adapting to changes in general daily routines, as reported by caregivers, which could further influence their ability to participate in structured school environments. While disability was not significantly associated with grade-for-age alignment, the findings suggest that conditional on attending school, children with disabilities progress at a similar pace to their peers, despite lower overall grade attainment.

The results align with prior research from LMICs showing that children with disabilities in low- and middle-income countries face multiple barriers to education, including financial constraints, health issues, inaccessible school infrastructure, and inadequate teacher training. [16,17] Children with disabilities in Nepal have lower school attendance rates, with a gap of about 2.7 percentage points compared to non-disabled children. These disparities mirror global patterns where disability remains one of the strongest predictors of school exclusion, often exceeding the impact of poverty or rural residence. [13,32] The findings also support earlier studies in Nepal that document how children with visual and hearing impairments, as well as those with physical disabilities, face systemic challenges in accessing education. [19] More recent evidence by Acharya and Yang in 2022 also finds that disability is strongly associated with lower school attendance and grade attainment, reinforcing the persistence of these educational disparities over time. [33]

In terms of academic progression, children with disabilities in this study were, on average, about 0.21 grades behind their peers after controlling for different socioeconomic factors. While this gap may seem small, early differences in grade attainment often widen over time, increasing the risk of repetition and dropout. [17] The fact that disability was not significantly associated with grade-for-age alignment suggests that once children with disabilities are enrolled, they tend to follow a similar progression as their peers. However, this does not necessarily mean they receive equal learning opportunities, as enrollment alone does not guarantee meaningful participation in school. [20]

Another key finding is the strong association between disability and adaptability. Children with disabilities were about 13.5 percentage points more likely to experience difficulty adjusting to routine changes, which could indicate broader

challenges in behavioral and psychological development. This aligns with studies showing that children with disabilities often struggle with school routines due to limited classroom accommodations, peer interactions, and teacher expectations. [34] These difficulties may not directly affect school enrollment but could influence classroom engagement, attendance, and long-term retention.

The interaction analyses (shown in Supporting Information S1 Table and S1 Fig) further indicate that the association between childhood disability and educational outcomes does not differ substantially by gender or area of residence. Although gender- and residence-based disparities in education are well documented in Nepal [30,31], the findings suggest that disability constitutes a cross-cutting disadvantage that affects children similarly across these groups. This pattern implies that the educational barriers faced by children with disabilities are not confined to specific demographic subpopulations but are instead systemic in nature [35].

There are some limitations in this study which needs to be noted. Firstly, the MICS dataset relies on caregiver-reported data, which may introduce reporting biases regarding both disability status and educational outcomes. Secondly, the dataset does not capture all dimensions of disability, such as psychosocial or neurological conditions, which may affect educational participation in different ways. Thirdly, the cross-sectional nature of the data allows to only find the relationship rather than causal effects. Lastly, while the study controls for multiple socioeconomic factors, unobserved influences such as school quality, individual learning differences, and household attitudes toward disability could still play major role in shaping the studied outcome variables.

This study focuses on estimating average associations between functional disability and educational outcomes. While we additionally examine heterogeneity in educational outcomes by gender and area of residence through interaction analyses, more finely disaggregated analyses—such as by disability type, severity, or socioeconomic status—could provide further insight into which subgroups of children with disabilities are most vulnerable. Future research using larger samples or more detailed and multidimensional measures of disability could build on these findings to explore subgroup-specific effects and inform more targeted policy interventions. Ideally, the research would combine large-scale survey data with qualitative insights from children, caregivers, and educators. As previous studies have suggested, qualitative work—particularly involving families and community perspectives—can uncover barriers that large-scale datasets may overlook [21,36].

## Conclusion

This paper contributes to the evidence base on disability and education in LMICs by using recent nationally representative data to examine how functional disability relates to key educational outcomes among children ages 5–17 in Nepal. The results confirm that children with disabilities face lower school attendance rates and attain fewer years of education on average than their peers after controlling for household, regional, and demographic factors. While progression through grades appears similar once enrolled, children with disabilities are more likely to struggle with behavioral and adaptive challenges that may limit their engagement in school.

In order to address the barriers children with disabilities face, policies must move beyond access and enrollment to focus on inclusive pedagogies, improved teacher training, and psychosocial support for children with disabilities. Investments in school infrastructure, data systems, and community engagement will also be essential to ensure that educational opportunities are both equitable and effective. While the study has limitations—particularly around the measurement of disability and the cross-sectional nature of the data—it provides a useful starting point for more targeted policy and research aimed at making education more inclusive for all children in Nepal.

## Supporting information

**S1 Table. Association between disability and educational outcomes by gender and area of residence.** Note: Values represent regression coefficients with standard errors in parentheses. Models include interaction terms between disability and gender (Panel A) and between disability and area of residence (Panel B). All models control for age group, household

characteristics, maternal education, and household wealth. Reference categories are male for gender and urban for residence. * $p < 0.10$, ** $p < 0.05$, *** $p < 0.01$.
(DOCX)

**S1 Fig. Association between disability and educational outcomes by gender and area of residence.** Note: Panel A presents the marginal effects of disability on four educational outcomes by gender, while Panel B presents the corresponding estimates by area of residence (urban versus rural). Outcomes include ever attending school, highest grade attained, age-appropriate grade enrollment, and difficulty adjusting to routine changes. Estimates are derived from linear regression models similar to the main analysis with interaction terms (between disability and the dimension of disadvantage) and adjusted for age group, household characteristics, maternal education, and wealth index.
(PDF)

## Acknowledgments

I prepared this manuscript when I was a student at the National College, Kathmandu University. I would like to thank UNICEF's MICS program for giving me access to the 2019 Nepal MICS data. I am grateful to Professor Yubraj Acharya at the Pennsylvania State University, United States, for providing comments on the previous version of this manuscript and offering to guide me through the publication process. All errors are my own.

## Author contributions

**Conceptualization:** Prashant Acharya.

**Data curation:** Prashant Acharya.

**Formal analysis:** Prashant Acharya.

**Investigation:** Prashant Acharya.

**Methodology:** Prashant Acharya.

**Resources:** Prashant Acharya.

**Software:** Prashant Acharya.

**Validation:** Prashant Acharya.

**Visualization:** Prashant Acharya.

**Writing – original draft:** Prashant Acharya.

**Writing – review & editing:** Prashant Acharya.

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
