## [Decision Letter · Decision Letter 0]

1 Dec 2025

PGPH-D-25-01937

Disability and Educational Outcomes in Nepal: Evidence from the 2019 Multiple Indicator Cluster Survey

Dear Dr. Acharya,

Thank you for submitting your manuscript to PLOS Global Public Health. After careful consideration, we feel that it has merit but does not fully meet PLOS Global Public Health’s publication criteria as it currently stands. Therefore, we invite you to submit a revised version of the manuscript that addresses the points raised during the review process.

We look forward to receiving your revised manuscript.

Kind regards,

Hassan Haghparast Bidgoli

Academic Editor

Journal Requirements:

Additional Editor Comments:

The reviewers have raised a number of concerns on your manuscript. Most of the issues raised relate to clarity in reporting, interpretation of results, and greater transparency in data management and variable definitions. I agree in particular with Reviewer 2’s suggestion to include additional contextual background on the Nepalese education system. I also support the recommendation to conduct subgroup or heterogeneity analyses where feasible, as these could substantially strengthen the contribution and policy implications of your research.

Please ensure that you address all comments raised by both reviewers in your revised manuscript and in the response letter.

Reviewers' comments:

Reviewer's Responses to Questions

**Comments to the Author**

1. Does this manuscript meet PLOS Global Public Health’s publication criteria? Is the manuscript technically sound, and do the data support the conclusions? The manuscript must describe methodologically and ethically rigorous research with conclusions that are appropriately drawn based on the data presented.

Reviewer #1: Yes

Reviewer #2: Yes

2. Has the statistical analysis been performed appropriately and rigorously?

Reviewer #1: Yes

Reviewer #2: Yes

3. Have the authors made all data underlying the findings in their manuscript fully available (please refer to the Data Availability Statement at the start of the manuscript PDF file)?

Reviewer #1: Yes

Reviewer #2: Yes

4. Is the manuscript presented in an intelligible fashion and written in standard English?

Reviewer #1: Yes

Reviewer #2: Yes

5. Review Comments to the Author

Reviewer #1: Thank you for the opportunity to review this important work on educational outcomes for students with disabilities. There are a few areas that could be clarified.

Abstract: Results – Last sentence: could you clarify what having fewer years of education is referring to – do you mean that more younger children had disability or that children with disability were less likely to have completed school? Also the statement “report facing challenges in accepting changes in their daily routine” could use clarification – what does this mean – is it referring to behavioral difficulty with transitions throughout the day?

Methodology: third paragraph, number (4) could you provide more detail about who is reporting difficulty with adjusting to routine changes (parent, teacher)

Table 1: In the two columns on the right, the percentages with disability and without disability for age, province, area and wealth are almost equal, this does not make sense. Could you provide more clarity for the reader to understand what these are referring to? It appears that up to 50% of children 5-9 years old have a disability for example.

Table 2: there is no information about which statistics you are reporting, I’d suggest using a standard table format that includes the statistics you are using in the header row under each variable. I also notice in the notes that you report p<0.1 – why are you reporting an insignificant value? Did you set your significance level at .1? Why did you make that decision?

Reviewer #2: I would like to thank the editor for the opportunity to review this manuscript. I would also like to thank the authors for covering this important topic of child disability and education outcomes in low- and middle-income countries. The following are my comments:

1. I think it would strengthen the manuscript to include a brief contextual overview of Nepal’s education system. For instance, at what age do children typically begin formal schooling? Which grades are covered under compulsory education? A concise paragraph covering the background would help readers unfamiliar with Nepal better situate the study within its national context.

2. Could you please explain what it means to have “difficulty adjusting to routine changes”? For example, are they routines at school, in study, or generally daily life routines?

3. Is there a particular reason why the authors divided the age groups as 5-9, 10-14, and 15-17?

4. While the analysis establishes a clear association between child disability and poorer educational outcomes, I think the study could make a better contribution to the literature with a more nuanced analysis, such as heterogeneity analysis. For example, the authors can examine whether the association differs by the type or severity of disability, gender, socioeconomic quintile, etc. Such a nuanced analysis could reveal which subgroups of children with disabilities are most vulnerable, providing much more targeted insights for policy. I offer this as a suggestion to enhance the paper's contribution, fully recognizing that the authors are the best judges of which specific analyses are most appropriate and feasible given their data and deep knowledge of the local context.

5. Following on the points 3&4, I think an alternative approach could be to divide the age groups according to educational stage—specifically, those during compulsory education and those beyond it. This distinction may reveal whether the negative association between child disability and education outcomes is attenuated during the compulsory schooling years.

6. PLOS authors have the option to publish the peer review history of their article (what does this mean?). If published, this will include your full peer review and any attached files.

**Do you want your identity to be public for this peer review?** For information about this choice, including consent withdrawal, please see our Privacy Policy.

Reviewer #1: No

Reviewer #2: No

Figure Resubmissions:

---

## [Decision Letter · Decision Letter 1]

4 Mar 2026

PGPH-D-25-01937R1

Disability and Educational Outcomes in Nepal: Evidence from the 2019 Multiple Indicator Cluster Survey

Dear Dr. Acharya,

Thank you for submitting your manuscript to PLOS Global Public Health. After careful consideration, we feel that it has merit but does not fully meet PLOS Global Public Health’s publication criteria as it currently stands. Therefore, we invite you to submit a revised version of the manuscript that addresses the points raised during the review process.

We look forward to receiving your revised manuscript.

Kind regards,

Hassan Haghparast Bidgoli

Academic Editor

Journal Requirements:

Additional Editor Comments (if provided):

Thanks for addressing the reviewers' concerns. Please see below for a few additional minor comments from the reviewer and incorporate them into your revised version.

Reviewers' comments:

Reviewer's Responses to Questions

**Comments to the Author**

1. If the authors have adequately addressed your comments raised in a previous round of review and you feel that this manuscript is now acceptable for publication, you may indicate that here to bypass the “Comments to the Author” section, enter your conflict of interest statement in the “Confidential to Editor” section, and submit your "Accept" recommendation.

Reviewer #1: All comments have been addressed

Reviewer #2: (No Response)

2. Does this manuscript meet PLOS Global Public Health’s publication criteria? Is the manuscript technically sound, and do the data support the conclusions? The manuscript must describe methodologically and ethically rigorous research with conclusions that are appropriately drawn based on the data presented.

Reviewer #1: Yes

Reviewer #2: Yes

3. Has the statistical analysis been performed appropriately and rigorously?

Reviewer #1: Yes

Reviewer #2: Yes

4. Have the authors made all data underlying the findings in their manuscript fully available (please refer to the Data Availability Statement at the start of the manuscript PDF file)?

Reviewer #1: Yes

Reviewer #2: Yes

5. Is the manuscript presented in an intelligible fashion and written in standard English?

Reviewer #1: Yes

Reviewer #2: Yes

6. Review Comments to the Author

Reviewer #1: Thank you for your responses, you addressed the comments made.

Reviewer #2: Thank you for thoroughly addressing my earlier comment; the revisions have strengthened the manuscript and improved the clarity of the results. My only remaining suggestion is about the additional results provided in the Supporting Information, which are currently presented as a figure. Given that these results are essentially regression coefficients if I understand correctly, I think it might be more interpretable and useful to readers if the regression coefficients are reported in a standard table format as well.

7. PLOS authors have the option to publish the peer review history of their article (what does this mean?). If published, this will include your full peer review and any attached files.

**Do you want your identity to be public for this peer review?** For information about this choice, including consent withdrawal, please see our Privacy Policy.

Reviewer #1: No

Reviewer #2: No

 Figure Resubmissions:

---

## [Decision Letter · Decision Letter 2]

18 May 2026

Disability and Educational Outcomes in Nepal: Evidence from the 2019 Multiple Indicator Cluster Survey

PGPH-D-25-01937R2

Dear Mr Acharya,

We are pleased to inform you that your manuscript 'Disability and Educational Outcomes in Nepal: Evidence from the 2019 Multiple Indicator Cluster Survey' has been provisionally accepted for publication in PLOS Global Public Health.

Best regards,

Julia Robinson

Executive Editor

Reviewer Comments (if any, and for reference):

Reviewer's Responses to Questions

**Comments to the Author**

1. If the authors have adequately addressed your comments raised in a previous round of review and you feel that this manuscript is now acceptable for publication, you may indicate that here to bypass the “Comments to the Author” section, enter your conflict of interest statement in the “Confidential to Editor” section, and submit your "Accept" recommendation.

Reviewer #2: All comments have been addressed

2. Does this manuscript meet PLOS Global Public Health’s publication criteria? Is the manuscript technically sound, and do the data support the conclusions? The manuscript must describe methodologically and ethically rigorous research with conclusions that are appropriately drawn based on the data presented.

Reviewer #2: Yes

3. Has the statistical analysis been performed appropriately and rigorously?

Reviewer #2: Yes

4. Have the authors made all data underlying the findings in their manuscript fully available (please refer to the Data Availability Statement at the start of the manuscript PDF file)?

Reviewer #2: Yes

5. Is the manuscript presented in an intelligible fashion and written in standard English?

Reviewer #2: Yes

6. Review Comments to the Author

Reviewer #2: The authors have addressed all my comments. Thank you!

7. PLOS authors have the option to publish the peer review history of their article (what does this mean?). If published, this will include your full peer review and any attached files.

**Do you want your identity to be public for this peer review?** For information about this choice, including consent withdrawal, please see our Privacy Policy.

Reviewer #2: No
